# Radionuclide Cisternography with [^64^Cu]Cu-DOTA

**DOI:** 10.3390/ph16091269

**Published:** 2023-09-07

**Authors:** Julia Greiser, Sebastian Groeber, Thomas Weisheit, Tobias Niksch, Matthias Schwab, Christian Senft, Christian Kuehnel, Robert Drescher, Martin Freesmeyer

**Affiliations:** 1Working Group for Translational Nuclear Medicine and Radiopharmacy, Nuclear Medicine, Jena University Hospital, 07747 Jena, Germany; julia.greiser@med.uni-jena.de; 2Nuclear Medicine, Jena University Hospital, 07747 Jena, Germanychristian.kuehnel@med.uni-jena.de (C.K.);; 3Neurology, Jena University Hospital, 07747 Jena, Germany; 4Neurosurgery, Jena University Hospital, 07747 Jena, Germany; christian.senft@med.uni-jena.de

**Keywords:** cisternography, radionuclide cisternography, cerebrospinal fluid leak, Cu-64 radiopharmaceuticals, Cu-64 DOTA, intrathecal injections, N,O ligands

## Abstract

Radionuclide cisternography (RNC) is a method for conducting imaging of the cerebrospinal system and can be used to identify cerebrospinal fluid leaks. So far, RNC has commonly employed radiopharmaceutical agents suitable only for single-photon emission tomography techniques, which are thus lacking in terms of image resolution and can potentially lead to false-negative results. Therefore, [^64^Cu]Cu-DOTA was investigated as an alternative radiopharmaceutical for RNC, employing positron emission tomography (PET) instead of single-photon emission tomography. A formulation of [^64^Cu]Cu-DOTA was produced according to the guidelines for good manufacturing practice. The product met the requirements of agents suitable for intrathecal application. [^64^Cu]Cu-DOTA was administered to a patient and compared to the approved scintigraphic RNC agent, [^111^In]In-DTPA. While no cerebrospinal fluid leak was detected with [^111^In]In-DTPA, [^64^Cu]Cu-DOTA RNC exhibited a posterolateral leak between the vertebral bodies C1 and C2. Thus, in this patient, PET RNC with [^64^Cu]Cu-DOTA was superior to RNC with [^111^In]In-DTPA. Since radiopharmaceuticals have a very good safety profile regarding the occurrence of adverse events, PET RNC with [^64^Cu]Cu-DOTA may become an attractive alternative to scintigraphic methods, and also to computed tomography or magnetic resonance imaging, which often require contrast agents, causing adverse events to occur much more frequently.

## 1. Introduction

The imaging of the cerebrospinal system (cisternography) is a diagnostic procedure intended to determine abnormalities and pathologies of the cerebrospinal fluid (CSF)-filled cavities, which include the subarachnoid space and the ventricular system around and inside the brain and spinal cord [1]. Pathologies are often caused by ruptures in the membrane surrounding the CSF cavities, e.g., due to trauma or surgery, and they are often accompanied by intracranial hypotension, headaches, nausea, or CSF leakage to the surrounding tissues, in some cases even from the nose or ear [2]. Unless surgical repair of the rupture can be performed, there is an increased risk of meningitis caused by unresolved CSF leakage [2]. Successful repair requires the preoperative identification and localization of the leakage site by means of suitable imaging methods.

Cisternography can be performed with various diagnostic imaging techniques, including high-resolution and/or contrast-enhanced computed tomography (CT) [3], contrast-enhanced or non-contrast-enhanced magnetic resonance imaging (MRI) [1,4], or radionuclide cisternography (RNC) by means of nuclear medical imaging methods [3,5,6,7,8]. Non-contrast-enhanced MRI is feasible as it is a non-invasive imaging technique, enabling a high-contrast-to-noise ratio, the multiplanar evaluation of volumetric data, and thin-section image acquisition. However, non-contrast-enhanced MRI only provides morphological information without a depiction of the actual CSF flow and thus has a considerable rate of false-negative results [1,2,4]. Furthermore, an MRI examination may be contraindicated, for example in patients with metal implants, like pacemakers. Similarly to MRIs, a non-contrast-enhanced high-resolution CT provides only anatomical information and thus only indirect evidence of CSF leakages or fistulas [2]. In these cases, the administration of either a CT or an MRI contrast agent (CA) into the CSF cavity by means of a lumbar puncture can help to identify the leak [2]. Similarly, radionuclide cisternography (RNC) requires the administration of a radiopharmaceutical into the CSF cavity, since the nuclear medical imaging modalities exclusively monitor the distribution of the radiopharmaceutical, although they are commonly combined with CT for anatomical reference and attenuation correction. Both CT and RNC examinations involve radiation exposure.

While every administration into the CSF cavity is invasive, contrast agents for CT and MRI are different from radiopharmaceuticals in terms of their chemical and pharmaceutical composition and the administered amounts. For example, for contrast-enhanced CT cisternography, an iodine containing agent is administered in high amounts of several hundred milligrams of iodine per milliliter [2,9]. Only one CT contrast agent, Iohexol (Omnipaque^TM^, GE Healthcare, Princeton, NJ, USA), is currently approved for intrathecal administration by the US Food and Drug administration (FDA) and the European Medicines Agency (EMA) and is generally considered to have a good safety profile, although reported adverse events include headaches, pain and aches, neck stiffness and nausea, and, albeit very rarely, seizures [9,10,11]. The overall adverse event incidence has been reported to be 7.4% to 40% [12]. According to the product information, an application of 4–10 mL of the Iohexol injection solution is recommended for cisternography, with a maximum total dose (10 mL) containing 3500 mg iodine or 7550 mg iohexol in Omnipaque 350^TM^ [11]. Similarly to CT contrast agents, MRI contrast agents contain high concentrations of the imaging agent and are based on gadolinium chelates. Gadolinium is a heavy metal, and intraventricular injections of high doses of gadolinium-based contrast agents may cause behavioral and neurological disturbances and histopathological changes [1]. A review into the safety of the intrathecal administration of gadolinium-based contrast agents, including more than 1000 patients, reported an overall rate of adverse events of 13%, while the incidence of serious adverse events, including serious neurotoxic complications (seizures, coma, and death) was 1% [12]. Serious adverse events were observed if the contrast agents were applied in doses greater than 1.0 mmol [12]. While, so far, no MRI contrast agent is approved for cisternography [4], the contrast agent most commonly used off-label for this purpose is gadopentetate dimeglumine (Magnevist^®^), which contains 0.5 mmol/mL of Gd-DTPA and is commonly administered in volumes of 0.5–1.0 mL [12,13].

In contrast to CT or MRI contrast agents, the concentration of excipients in radiopharmaceutical formulations is very low, thus ensuring the high-quality safety profiles of radiopharmaceuticals in general [14]. Cases of adverse events reported for the intrathecal administration of radiopharmaceuticals could be explained by calcium(II) and magnesium(II) ion sequestration caused by the presence of excess chelating agent in some formulations, which by now have been identified as being unsuitable for intrathecal administration [15]. Therefore, an equimolar amount of calcium(II) ions must always be contained in the radiopharmaceutical formulation intended for intrathecal administration [15].

Conventional RNC is performed with planar scintigraphy and single-photon emission computed tomography (SPECT). Two different radiopharmaceuticals are commonly used for RNC: [^111^In]In-DTPA and [^99m^Tc]Tc-DTPA. To date, [^111^In]In-DTPA is the only radiopharmaceutical approved by the EMA for intrathecal use (In DTPA Injection, Curium (Mallinckrodt), Petten, The Netherlands; and Indium DTPA In111, GE Healthcare Inc., Mississauga, ON, USA) [16,17,18]. RNC with [^111^In]In-DTPA has been reported to be reliable for detecting CSF leaks [6,19]. Since the imaging modality exclusively detects signals stemming from the radiopharmaceutical that is distributed within the intracranial cavity, while at the same time being combined with a CT to provide an anatomical reference and attenuation correction, RNC is a highly selective imaging method. However, the main limitations of RNC when employing the SPECT technique and using [^111^In]In-DTPA are the limited image quality and low spatial resolution, which ultimately lead to RNC often not being favored as the modality of choice in comparison to CT and MRI [1,2,5]. The low image resolution is caused by the high photon energy of the indium-111 radionuclide [20]. For this reason, [^99m^Tc]Tc-DTPA has emerged as a feasible alternative to [^111^In]In-DTPA, providing superior image quality due to the lower photon energy of technetium-99m [20]. For intrathecal administration, [^99m^Tc]Tc-DTPA is prepared from preservative-free calcium trisodium DTPA kits [21,22].

Regardless of the applied radionuclide, RNC with SPECT remains inferior to its ‘sister’ imaging technology, which is positron emission tomography (PET). Compared to SPECT, PET exhibits higher sensitivity and better image quality and spatial resolution, and it combines three-dimensional acquisition with high temporal resolution, which makes it feasible for use in the imaging of dynamic processes and the identification of small structures. Therefore, we sought to develop a radiopharmaceutical suitable for PET radionuclide cisternography.

The two most common PET radionuclides, fluorine-18 (half-life 110 min) and gallium-68 (half-life 68 min), are rather short-lived. So far, a study of [^68^Ga-]Ga-DOTA has produced promising results for CSF leak detection with PET RNC [23]. Additionally, an ^18^F-labeled fluorescein was successfully used for CSF imaging in rats [24]. Due to the slow CSF circulation dynamics, it takes several hours until the full homogeneous distribution of an agent within the CSF cavity is achieved. Therefore, a longer-lived PET nuclide was deemed favorable for PET RNC. With a half-life of 12.8 h, the PET radionuclide copper-64 is often used for the imaging of slow processes, e.g., via antibody labeling [25]; thus, it may also be suitable for cisternography intended to localize CSF leaks. Furthermore, compared to short-lived radionuclides like gallium-68, a cisternography agent based on copper-64 would allow for centralized production and long-range distribution.

To prevent preliminary diffusion from the CSF cavity into the blood via the blood–CSF barrier [26], the [^64^Cu]copper(II) ion needs to be chelated by a ligand, similarly to the chelation of the radionuclides in [^111^In]In-DTPA and [^99m^Tc]Tc-DTPA by DTPA. The most suitable and commonly used chelators for copper-64 are macrocycles like DOTA (1,4,7,10-Tetraazacyclododecane-1,4,7,10-tetraacetic acid), TETA (1,4,8,11-tetraazacyclotetradecane-N,N′,N″,N″′-tetraacetic acid), and cyclam (1,4,8,11-tetraazacyclotetradecane) [25,27,28]. Recently, the use of [^64^Cu]Cu-albumin for CSF imaging and the evaluation of lymphatic efflux in mice was also reported [29,30]. As it is commercially available and due to FDA administration approval [28], we chose the DOTA ligand as the chelator for the copper-64 radionuclide in the formulation. In 2019, the first successful CSF leak detection with PET radionuclide cisternography using [^64^Cu]Cu-DOTA was reported [31]. Herein, we describe in detail the radiopharmaceutical production of [^64^Cu]Cu-DOTA in compliance with standards of good manufacturing practice (GMP) and its use in a clinical case, where PET RNC with [^64^Cu]Cu-DOTA proved superior to conventional RNC with [^111^In]In-DTPA.

## 2. Results

### 2.1. Small- and Full-Scale Test Batch Production

The radiolabeling efficiency of [^64^Cu]Cu-DOTA could be raised by increasing the DOTA precursor amount (Figure 1). For the quantitative labeling of 3.5–4.5 MBq [^64^Cu]Cu-chloride in PBS at pH 7.3–7.4, in a total volume of 550 µL, a DOTA amount of 0.4 µg was sufficient, equaling a concentration of about 0.7 µg/mL DOTA. This means that, in a full batch production containing 60 ± 5 MBq [^64^Cu]Cu-chloride starting activity, 6 µg DOTA should be sufficient to ensure quantitative labeling.

Following small-scale production, four [^64^Cu]Cu-DOTA batches mimicking the conditions of a full scale production were produced, using different starting amounts of DOTA in a total reaction volume of 5 mL PBS. The starting activities of [^64^Cu]Cu-chloride were kept constant at 60 ± 5 MBq. Unlike the theoretical amount of DOTA (6 µg) necessary for quantitative radiolabeling (defined as RCP ≥ 95.0%), which was determined by extrapolating the results from the small-scale productions, we found that, for the successful radiolabeling of 60 ± 5 MBq [^64^Cu]Cu-chloride, more than 10 µg of DOTA were required (Table 1). Batches nos. 3 and 4, containing 20 µg DOTA, yielded [^64^Cu]Cu-DOTA in RCP ≥ 95.0%, while batches nos. 1 and 2, which were produced with 2 and 10 µg DOTA, respectively, contained [^64^Cu]Cu-DOTA in unsatisfactory RCP (Table 1), indicating precursor amounts ≤ 10 µg to be insufficient for successful radiolabeling. These differences in the precursor amount between small-scale and full-scale production might be explained by upscaling effects (e.g., larger volumes and a higher total amount of non-radioactive impurities) and the change of the reaction vessel from a polypropylene reaction tube to a glass vial.

Of these four full-scale test batches with starting activities of 60 ± 5 MBq, three batches (nos. 1–3) were produced on the day of delivery, requiring 120 µL each of undiluted [^64^Cu]Cu-chloride (used as delivered) as the starting amount. A fourth batch (no. 4) was produced 24 h after the first three batches. Due to the radioactive decay of ^64^Cu, for the fourth batch, a volume of 400 µL of [^64^Cu]Cu-chloride was necessary to reach the required starting activity of 60 ± 5 MBq. This resulted in a notably lower pH value of the reaction solution no. 4 (pH 6.8 compared to 7.3–7.4 in nos. 1–3), which did not impair the RCP of [^64^Cu]Cu-DOTA (Table 1). If the [^64^Cu]Cu-DOTA preparation takes place on the day of delivery around noon, about 18 h have passed since the [^64^Cu]Cu-chloride precursor solution was produced at Biont a.s. (Bratislava, Slovakia). If the [^64^Cu]Cu-DOTA is prepared on the day after delivery, 42 h have passed since radionuclide precursor production, which is still within the expiration time (48 h) of the [^64^Cu]Cu-chloride. The test batch production results indicate that [^64^Cu]Cu-DOTA can be produced successfully even when using [^64^Cu]Cu-chloride solution up to 42 h after production. These findings were confirmed by the GMP batch productions, both on the day of delivery and 24 h after delivery (see Section 2.3).

### 2.2. Stability of [^64^Cu]Cu-DOTA

Test batch no. 3 was used to test [^64^Cu]Cu-DOTA stability in human serum (37 °C), in cerebrospinal fluid (37 °C) and against a 500-fold excess of calcium(II) chloride, respectively. In all three samples, no demetallation was observed with radio TLC and radio HPLC after 24 h, indicating the sufficiently high stability of the [^64^Cu]Cu-DOTA complex.

### 2.3. GMP Production of [^64^Cu]Cu-DOTA

In total, 12 batches of [^64^Cu]Cu-DOTA were produced, containing about 60 MBq in 10 mL. Since no purification steps except for sterile filtration are necessary during the production, the RCY were ≥90% (Table 1). Moreover, 50 µg DOTA were used as a starting amount in all batches, meaning that the final formulations had a specific activity of 1.2 ± 0.1 MBq/µg. Radiolabeling was performed at 100 °C for 15 min, yielding batches of RCP ≥ 90%. We found that increasing the reaction time to 25 min instead of 15 min did not increase the RCP any further. There was no out-of-specification event.

Radio HPLC of the first four batches was initially performed with a gradient consisting of water and acetonitrile and containing trifluoroacetic acid (0.1%), instead of the buffered HPLC gradient that was used for the analysis of the further eight batches (see Section 4.7). However, the initial HPLC gradient was identified to be unsuitable for analysis, due to partial decomposition of [^64^Cu]Cu-DOTA at the pH of the gradient system. Therefore, the RCP (HPLC) is only given for batches nos. 5–12 (n = 8, Table 2).

The shelf-life of [^64^Cu]Cu-DOTA was determined via radio TLC and radio HPLC analyses of the produced batches 24 h and 48 h after production. The RCP at 24 h and 48 h remained at ≥90%, indicating the high stability of the [^64^Cu]Cu-DOTA formulation, allowing for application up to one day after production. Accordingly, the expiration time was defined to be 24 h after production.

### 2.4. Quality Control of [^64^Cu]Cu-DOTA

All batches of [^64^Cu]Cu-DOTA complied with the specifications on sterile and pyrogen-free injection solutions, as required by the European Pharmacopeia (Ph. Eur.). The determination of endotoxin content and sterility testing were performed according to the methods described in the Ph. Eur (Table 2) [32,33]. The endotoxin content of all batches was ≤0.5 IU/mL and all batches were sterile. The pH value and osmolarity were within the acceptable limits for intrathecal injections, as they were in accordance with the pH and osmolarity of healthy CSF fluid (Table 2) [34].

As yet, there is no published monograph on the ^64^Cu precursor solution or ^64^Cu radiopharmaceuticals. Thus, the limit for radionuclidic purity (RNP) was taken from the certificate of analysis of the delivered [^64^Cu]Cu-chloride solution. We considered an RCP ≥ 90.0% sufficient for image quality, since [^64^Cu]Cu-DOTA is not a specific targeting substance; instead, it distributes homogeneously but unspecifically within the CSF cavity. A content of non-chelated [^64^Cu]Cu-chloride substantially higher than 10.0%, however, may lead to the diffusion of non-negligible amounts of activity from the CSF cavity, thus impacting diagnostic accuracy. The RCP of [^64^Cu]Cu-DOTA could be determined using radio HPLC and radio TLC. Both methods were suitable for quantifying the percentage of [^64^Cu]Cu-chloride and thus the RCP of [^64^Cu]Cu-DOTA. Peak separation between [^64^Cu]Cu-chloride and [^64^Cu]Cu-DOTA was sufficient in both the radio TLC and radio HPLC analyses (Figure 2 and Figure 3).

### 2.5. Pharmaceutical Composition of the [^64^Cu]Cu-DOTA Formulation

It is vital to reflect on all excipients present in the [^64^Cu]Cu-DOTA formulation to reduce the probability of adverse events to the greatest extent possible. Naturally, the amount of the DOTA precursor is much higher than the amount of the radiopharmaceutical [^64^Cu]Cu-DOTA. Since 50 µg (0.12 µmol) of DOTA is used as the starting amount and since there are no additional purification steps suitable for reducing the precursor content in the formulation, the amount of DOTA in the final formulation of [^64^Cu]Cu-DOTA must be assumed to be 50 µg in 10 mL, equaling 25 µg or 0.06 µmol per administration volume (5 mL) (Table 3). This represents a low molar amount, compared to the molar amounts of the DTPA precursor used in commercially available [^111^In]In-DTPA and in various [^99m^Tc]Tc-DTPA preparations, which are considered suitable for intrathecal administration [15]. [^111^In]In-DTPA is available from two commercial manufacturers; Curium and GE Healthcare Inc. While 0.5–1 mL of [^111^In]In-DTPA (commonly used for RNC) contains 0.13–0.25 µmol of DTPA, formulations of [^99m^Tc]Tc-DTPA have been reported to contain up to 1 mg (2 µmol) of DTPA per administration in the chemical form of CaNa_3_DTPA; meanwhile, the content may vary, depending on which kit is used for [^99m^Tc]Tc-DTPA preparation and the volume that is injected (Table 3) [15].

Both [^64^Cu]Cu-DOTA and [^111^In]In-DTPA (Curium) are buffered using PBS buffer, exhibiting a final pH of 7.0–7.6, while [^111^In]In-DTPA (GE Healthcare) is buffered using a sodium bicarbonate buffer (similar to physiological CSF), reaching a comparable pH range of 7.0–8.0. In contrast, [^99m^Tc]Tc-DTPA preparations from a kit commonly use saline for reconstitution, which results in pH values of 4–5, so that additional buffering using sodium hydroxide may be required [15]. The content of phosphate ions and sodium chloride is higher per administration of [^64^Cu]Cu-DOTA compared to [^111^In]In-DTPA by factor 5–6.5, which is in accordance with the higher administered volume of [^64^Cu]Cu-DOTA (5 mL) compared to [^111^In]In-DTPA (1 mL).

[^99m^Tc]Tc-DTPA preparations require the presence of stannous(IV) chloride as reducing agent and thus always contain stannous(II) chloride or stannous(II)-DTPA, in some cases alongside antioxidants like gentisic acid or sodium *p*-amino-benzoic acid [15]. In contrast, [^64^Cu]Cu-DOTA and [^111^In]In-DTPA formulations contain no additional reducing agents or antioxidants.

### 2.6. Biodistribution, Diagnostic Findings, and Excretion Profile of [^64^Cu]Cu-DOTA

PET RNC with [^64^Cu]Cu-DOTA was performed in a patient suspected of having a CSF leak due to experiencing symptoms of chronic CSF loss. An initial unenhanced MRI prior to the RNC had been inconclusive. Similarly, a SPECT RNC using [^111^In]In-DTPA did not reveal any indication of a CSF leak since no activity was detected outside of the CSF cavity (Figure 4a). As expected, [^111^In]In-DTPA distributed around the brain stem at two hours p.i., and around the cerebellum and temporal convexities and in the fourth ventricle six hours p.i (Figure 4a). A late SPECT scan 22 h p.i. showed tracer activity over the cerebral convexities as well as in the third and fourth ventricles. At no time point during image acquisition, a leak was detected.

In view of the severity of the clinical symptoms and the possibility of false-negative results using the established modalities, RNC was repeated in the same patient one day later, using [^64^Cu]Cu-DOTA. At 40 min p.i. the radiotracer distribution had reached the brain stem. At 2 h p.i., substantial tracer activity was detected at the skull base and in the fourth ventricle. Additionally, tracer activity in the third ventricle and over the cerebral convexities was slightly visible at two hours p.i. and clearly visible at six hours p.i. At all acquisition time points, a left posterolateral leak between vertebral bodies C1 and C2 (atlas and axis) was observed (Figure 4b and Figure 5). Additionally, activity was seen in the sleeve of the left C5 nerve root sleeve (Figure 4b), which may represent a normal variant.

Urine samples were collected at different time points after the administration of the tracer. Samples were taken at 16.5 h and 17.5 h p.i. in the morning after an overnight stay and exhibited the highest activity concentration. Renal excretion continued with decreasing activity concentration until, in total, 22 MBq (i.e., >70% of the total injected activity) had been excreted via the urinary tract after 24 h p.i. (Figure 6, decay corrected). Accordingly, the PET/CT scans showed activity in the bladder, as well as low levels of activity in the liver.

## 3. Discussion

The evaluation of CSF leaks can be performed with various diagnostic imaging techniques, including CT, MRI, and radionuclide cisternography. The novel PET RNC procedure with [^64^Cu]Cu-DOTA was proven to be suitable for the identification of CSF leaks and to be superior regarding image resolution compared to conventional SPECT.

The production and quality control of [^64^Cu]Cu-DOTA was performed with regard to GMP requirements, using a GMP-certified, commercially available radionuclide precursor [^64^Cu]Cu-chloride, while further components (DOTA, PBS buffer, and calcium(II) chloride dihydrate) were acquired at chemical or pharmaceutical grade. The radiolabeling of DOTA with [^64^Cu]Cu-chloride (60 MBq) was performed manually, using 50 µg of DOTA. Although 20 µg of the DOTA precursor was shown to be sufficient for the quantitative radiolabeling of [^64^Cu]Cu-chloride (60 MBq), 50 µg of DOTA was established as the starting amount for the GMP production batches, in order to ensure quantitative labeling with high certainty.

[^64^Cu]Cu-DOTA was produced in high RCY (≥90%) and RCP (≥90.0%). The PBS buffer exhibited sufficient capacity to buffer up to 400 µL of a [^64^Cu]Cu-chloride precursor solution (pH 0.7–1.4), providing a final [^64^Cu]Cu-DOTA product pH of 7.0–7.6. This is relevant because, if a [^64^Cu]Cu-chloride solution with a lower pH or distinctly higher volume is used, the resulting pH of the final formulation may fall below the specified lower pH limit of 7.0. Thus, we recommend calculating the proton concentration of the [^64^Cu]Cu-chloride solution prior to production to ensure that the pH will be within the specifications.

[^64^Cu]Cu-DOTA exhibited high in vitro stability, which is in accordance to a previously reported study, although another group observed comparably fast transchelation [27,35]. In any case, for successful RNC diagnosis, it is sufficient that [^64^Cu]Cu-DOTA exhibits high stability in the practically protein-free CSF, thus preventing preliminary excretion of ions from the CSF cavity. Potential transchelation, occurring once the tracer has crossed the CSF–blood barrier, is then negligible.

Due to the direct injection into the CSF cavity, administrations intended for intrathecal use must comply with particularly strict specifications regarding the level of possible impurities and microbial contamination. To ensure sterility and sufficiently low endotoxin levels, all requirements for aseptic production, including production under GMP grade A, the use of sterile compounds suitable for pharmaceutical production whenever applicable, and sterile filtration, were considered and executed with the utmost care. The US Pharmacopeia (USP) proscribes a maximum of ≤14.0 IU/V bacterial endotoxins for radiopharmaceuticals that are administered via intrathecal injection [21,36]. Given the total volume of a produced batch of [^64^Cu]Cu-DOTA is 10 mL, whereof a maximum of 5 mL is administered per patient, this would set the acceptance level at ≤2.8 IU/mL per application. However, for means of risk reduction, we chose ≤1.0 IU/mL as the acceptance level and all batches complied with this specification.

The presence of excipients that may exhibit pharmacological activity must be strictly avoided whenever possible. That includes excess chelator and unnecessary anti-oxidants or reducing agents, which may be present in [^99m^Tc]Tc-DTPA formulations [15]. The addition of calcium(II) chloride in equimolar amounts to the chelator is crucial to ensure that the excess chelator in the final formulation does not lead to the sequestration of calcium(II) or magnesium(II), as has been reported previously for some formulations of [^99m^Tc]Tc-DTPA [15]. The sequestration of calcium(II) or magnesium(II) may lead to irreversible blocking of nerve conduction and has reportedly resulted in adverse events [15]. Surprisingly, we found that, in contrast to [^111^In]In-DTPA (Curium), the [^111^In]In-DTPA manufactured by GE Healthcare does not seem to contain any Ca(II), according to the product leaflet [16,17]. However, adverse events listed for [^111^In]In-DTPA (GE Healthcare), like aseptic meningitis and pyrogenic reactions, exhibit an incidence of less than 0.4% and may also be related to the lumbar puncture procedure itself, rather than to the radiopharmaceutical [16]. In healthy humans, the CSF contains about 1.2 mmol/L of Ca(II) ions in a total volume of roughly 150 mL, equaling a total molar amount of 180 µmol of Ca(II) ions in the CSF [37]. Thus, less than 0.1% of the Ca(II) ions in the CSF will be sequestrated by the excess DTPA (0.13 µmol) present in the [^111^In]In-DTPA produced by GE Healthcare, which may explain the low incidence of adverse events.

Compared to [^111^In]In-DTPA, the molar amounts of excess DOTA chelator (0.06 µmol per administration) present in the formulation of [^64^Cu]Cu-DOTA as described herein are even lower and are accompanied by an equimolar amount of calcium(II) chloride. In contrast, some [^99m^Tc]Tc-DTPA formulations, which have been identified as unsuitable for intrathecal administration, contain up to 24 mg (61 µmol) of excess DTPA [15] and may thus sequestrate up to a third of the Ca(II) ions contained in the CSF, resulting in notable adverse events. Therefore, among the three known radiopharmaceutical formulations suitable for RNC, [^64^Cu]Cu-DOTA is the one least likely to cause any pharmacological adverse events caused by excess chelator. Furthermore, formulations of [^99m^Tc]Tc-DTPA by necessity contain stannous(II) chloride or stannous(II)-DTPA, which may also affect nerve function [15], while [^64^Cu]Cu-DOTA and [^111^In]In-DTPA formulations are free of stannous(II) reagents, since ^64^Cu and ^111^In labeling do not require reducing agents.

Compared to these three RNC radiopharmaceuticals, the contrast agents used for cisternography with CT (Iohexol) or MRI (Gd-DTPA) contain high concentrations of the imaging substance. Omnipaque^®^ contains 0.37 mmol/mL of Iohexol, Magnevist^®^ contains 0.5 mmol/mL of Gd-DTPA. Thus, both contrast agents carry a significant risk of adverse events. In contrast, our formulation of [^64^Cu]Cu-DOTA contains DOTA and calcium(II) chloride in concentrations of 0.012 µmol/mL (or 0.06 µmol per administration), meaning a dose that is smaller by a factor of about 1000.

The natural buffer system within the CSF is a carbonate buffer, which keeps the CSF pH constant within the range of 7.32–7.36 [34,38]. The PBS buffer system used for [^64^Cu]Cu-DOTA is similar to the one used for [^111^In]In-DTPA (Curium). Although a PBS buffer of a higher concentration would exhibit a higher capacity for buffering [^64^Cu]Cu-chloride solutions of excessive acidity and volume, we chose not to exceed a PBS concentration of 0.02 mmol/mL (meaning a total molar amount of phosphate 0.09 mmol per administration) in order to avoid negative influences on the CSF osmolarity and ion concentration.

No adverse events were observed following the intrathecal application of [^64^Cu]Cu-DOTA. We demonstrated that the distinctly improved image resolution of the PET modality led to the identification of a CSF leak, while SPECT RNC with [^111^In]In-DTPA yielded a false-negative result. The intracranial distribution of [^64^Cu]Cu-DOTA in the CSF space was faster than that of [^111^In]In-DTPA; the tracers reached the skull base, cerebellum, and fourth ventricle two hours and six hours after the lumbar injection, respectively. Distribution over the cerebral convexities was seen on [^64^Cu]Cu-DOTA PET RNC after 6 h and on [^111^In]In-DTPA SPECT RNC only in the next-day image (22 h). The cervical CSF leak was already apparent on the [^64^Cu]Cu-DOTA PET RNC images acquired 40 min after injection but was best seen after 2 h and 6 h.

CSF fluid is produced at a rate of about 500 mL/day, meaning that the entire CSF volume (150–170 mL) is renewed every 5–7 h [38]. The main portion of the CSF volume is reabsorbed by the tissues of the brain, while a smaller part portion into venous sinuses and lymphatic vessels [39]. Substances that are administered via intrathecal injection can be expected to leave the intracranial cavity over time and be excreted via the blood stream and subsequent renal and/or hepatobiliary pathways. In total, more than 70% of the total injected activity of [^64^Cu]Cu-DOTA was found in the urine, indicating renal excretion as the major route. This is comparable to [^111^In]In-DTPA, wherein two third of the radiopharmaceutical activity are cleared via the kidneys [40], while 90% of Iohexol are cleared through the kidneys [9].

The single case reported herein serves as a proof-of-concept of the superiority of [^64^Cu]Cu-DOTA PET RNC over [^111^In]In-DTPA SPECT RNC. To further evaluate the potential of [^64^Cu]Cu-DOTA, not just for CSF leak detection but also for studying further abnormalities and fluid characteristics, studies on a larger number of patients are of course required first. The authors plan to undertake these studies in the near future.

## 4. Materials and Methods

### 4.1. [^64^Cu]Cu-Chloride Solution

The [^64^Cu]Cu-chloride solution was purchased as a radiopharmaceutical precursor of GMP grade from BIONT, a.s., Bratislava, Slovakia. [^64^Cu]Cu-chloride was delivered as a clear, colorless solution in a conical glass vial, exhibiting a pH range of 0.7–1.4, a radionuclide purity of ≥99.5%, and a radiochemical purity of ≥95.0%.The delivered batches of [^64^Cu]Cu-chloride solution exhibited varying activity concentrations between 583–901 MBq/mL; therefore, the volume required for [^64^Cu]Cu-DOTA production was calculated prior to synthesis.

### 4.2. Preparation of PBS Buffer, Calcium(II) Chloride, and DOTA Stock Solution

All solutions were prepared in a biosafety cabinet (GMP grade A). Weighing of substances was performed using sterile plastic spatulas to avoid metal contamination. The PBS buffer (pH 7.7–7.9) was prepared by dissolving 298 mg disodium hydrogen phosphate (EMSURE^®^ Reag. Ph Eur, Merck KGaA, Darmstadt, Germany), 25 mg potassium dihydrogen phosphate (EMSURE^®^ Reag. Ph Eur, Merck KGaA, Darmstadt, Germany), and 1.0 g of sodium chloride (99.99 Suprapur^®^, Merck KGaA, Darmstadt, Germany) in 125 mL of sterile water of ultrapure grade (ULTREX™, J. T. BakerTM, Avantor™ Performance Materials, Center Valley, PA, USA).

The DOTA stock solution (5 mg/mL) was prepared by dissolving 50 mg of DOTA (Sigma Aldrich, Merck KGaA, Darmstadt, Germany, purity (CHN) ≥ 97.0%) in 10 mL water of ultrapure grade. Calcium(II) chloride solution (0.75 mg/mL) was prepared by dissolving 12 mg calcium(II) chloride dihydrate (tested according to Ph. Eur., Merck KGaA, Darmstadt, Germany) in 16 mL of sterile ultrapure water.

All stock solutions were freshly prepared on the day of production. For batches produced 24 h after delivery, the solutions were kept overnight in the fridge (4–8 °C).

### 4.3. Small-Scale Test Batch Production for the Determination of Radiolabeling Efficiency

A stock solution (0.01 mg/mL) of DOTA was prepared using ultrapure water. To an aliquot of buffer (500 µL) in a reaction tube was added a respective volume (0–50 µL) of the DOTA stock solution, containing 0–0.5 µg. Subsequently, the relevant volume of water (50–0 µL) was added to keep the total volume of the reaction mixture at 550 µL. The buffered solution was homogenized by brief shaking. Subsequently, the reaction tube was placed in a lead shield and an aliquot of 8 µL of [^64^Cu]Cu-chloride solution per tube, containing 3.5–4.5 MBq at time of labeling, was added to the buffered solution. Activities were determined using a dose calibrator (ISOMED 2010, NUVIA Instruments, Dresden, Germany). The reaction tube was placed in a heating block (100 °C, Ministir II, Zinsser Analytics GmbH, Frankfurt, Germany) for 15 min. Subsequently, the reaction was allowed to cool to room temperature and a sample (5 µL) was withdrawn for radio TLC analysis. The experiment was performed in triplicates for each defined amount of DOTA stock solution.

### 4.4. Full-Scale Test Batch Production for the Determination of the Precursor Amount

Four batches mimicking the conditions of a full-scale labeling reaction were produced, using different starting amounts of DOTA, in order to determine the necessary precursor amount. A sterile evacuated glass vial sealed with a rubber cap (Mallinckrodt) was placed in a lead shield. PBS buffer (2 mL) was placed in a sterile reaction tube with a conical bottom. From a stock solution (1 mg/mL) of DOTA in 2 µL (2 µg) of ultrapure water, 10 µL (10 µg) or 20 µL (20 µg) was removed and mixed with the PBS. Subsequently, the solution was withdrawn with a syringe, equipped with a metal cannula, and transferred into the glass vial. An equal volume of air was withdrawn with the syringe, to keep the air pressure within the glass vial constant.

Then, 3 mL of PBS buffer was placed into a second sterile reaction tube. Using a pipette with sterile pipette tips, an aliquot of the [^64^Cu]Cu-chloride solution was withdrawn from the delivery vial and transferred into the PBS solution. For test batches nos. 1, 2, and 3, which were produced on the day of delivery, 120 µL (60 ± 5 MBq) of [^64^Cu]Cu-chloride was removed. For test batch no. 4, produced the day after delivery, 400 µL (60 ± 5 MBq) of [^64^Cu]Cu-chloride was removed. Using a syringe equipped with a metal cannula, the buffered [^64^Cu]Cu-chloride solution was transferred into the glass vial. The glass vial was placed in a heating block (100 °C, Ministir II, Zinsser Analytics GmbH Frankfurt, Germany) for 15 min. Samples for RCP analysis were withdrawn after letting the vial cool to room temperature.

### 4.5. Stability Determination

From test batch no. 3, an aliquot of 100 µL was removed and mixed with 400 µL of human serum (from human male AB plasma, USA origin, sterile-filtered, Merck KGaA, Darmstadt, Germany). Another aliquot of 100 µL of test batch no. 3 was mixed with 400 µL of fresh cerebrospinal fluid taken from a patient. The samples were incubated at 37 °C and aliquots for radio TLC and HPLC analyses were removed after 12 h and 24 h. The sample incubated with human serum was centrifuged (3500 rpm, 685 rcf, 5 min) prior to the HPLC analysis, in order to remove insoluble parts.

Another aliquot (1000 µL, equaling a content of 4 µg or 0.01 µmol of DOTA precursor) of test batch no. 3 was mixed with 100 µL (0.51 µmol) of calcium(II) chloride solution (0.75 mg/mL), resulting in an excess of Calcium(II):DOTA of 500:1; then, it was kept at room temperature. Aliquots for the radio TLC and radio HPLC analyses were removed after 12 h and 24 h.

### 4.6. GMP Production

The radiolabeling was performed manually in a biosafety cabinet (GMP grade A). In preparation, a sterile evacuated glass vial sealed with a rubber cap (Mallinckrodt) and two sterile reaction tubes with conical bottoms (5 mL) were placed in lead shields. Then, 3 mL of PBS buffer was placed into the first sterile reaction tube, and 2 mL of PBS buffer was placed into the second sterile reaction tube. Using an autoclaved pipette with sterile pipette tips, an aliquot of [^64^Cu]Cu-chloride solution (65–250 µL, 60 ± 5 MBq) was withdrawn from the delivery vial and transferred into the first sterile reaction tube containing 3 mL of PBS solution. In the second sterile reaction tube, an aliquot (10 µL) of DOTA stock solution (5 mg/mL) was mixed with the PBS (2 mL). Then, both the [^64^Cu]Cu-chloride/PBS solution and, subsequently, the DOTA/PBS solution were withdrawn from the reaction tubes, using a syringe (10 mL) equipped with a metal cannula. The combined solutions were then transferred into the sterile evacuated glass vial, followed by the removal of an aliquot of air (5 mL) to keep the pressure within the vial constant. The glass vial was placed in a heating block (100 °C, Ministir II, Zinsser Analytics GmbH Frankfurt, Germany) for 15 min.

During the heating of the reaction, 5 mL of PBS buffer was placed into a third sterile reaction tube with conical bottom (5 mL). Then, 12 µL of calcium(II) chloride solution (0.75 mg/mL) was transferred into the PBS using an autoclaved pipette with sterile pipette tips.

After letting the reaction cool to room temperature, the entire volume of the vial (5 mL) was withdrawn using a 20 mL syringe and a metal cannula. A second sterile evacuated vial, which had been equipped with a short cannula and a vented sterile filter (0.22 µm) and a long cannula and a vented sterile filter (0.22 µm), was placed in a lead shield. The entire volume of the [^64^Cu]Cu-DOTA formulation was slowly passed through the sterile filter on the long cannula into the vial. Then, using the same syringe and metal cannula, the calcium(II) chloride/PBS solution was withdrawn from the reaction tube. The syringe was placed in the same sterile filter and the solution was slowly passed through the filter into the vial. Afterwards, the vial was briefly shaken to ensure homogenization.

The vented filter was removed from the long cannula and a sample (2 mL) was withdrawn for quality control.

### 4.7. Quality Control

Radiochemical purity (RCP) was determined via radio HPLC using a eurosphere 125 × 4 mm column (100–5 C18, Knauer) and the following gradient: 0.0–2.0 min 100% B, 2.0–5.0 min 100% B → 80% B, 5.0–7.0 min 80% B → 65% B, 7.0–9.0 min 65% B → 0% B, 9.0–11.0 min 100% A, 11.0–11.05 min 0.0% B → 100% B, 11.05–13.00 min 100% B, with A being acetonitrile and B being triethylammonium phosphate solution (50 mM, pH 3.6; prepared from mixing 855 µL orthophosphoric acid, 1.26 g trimethylamine in 250 mL of water). This method was established in the same way as a previously published protocol [35].

Additionally, radio TLC was performed using silica gel plates on aluminum (1 cm × 6 cm Merck, Darmstadt, Germany) and an eluent of methanol/ammonium acetate (77 g/L) (1/1 *v*/*v*).

### 4.8. [^111^In]In-DTPA and [^64^Cu]Cu-DOTA Administration and Imaging Protocol

Injection solutions of [^111^In]In-DTPA (1 mL, 37 MBq) and [^64^Cu]Cu-DOTA (5 mL, 30 MBq) were administered to the patient on separate days via lumbar puncture between vertebrae L2 and L3. Following the [^111^In]In-DTPA administration, SPECT acquisitions were performed at 2 h, 6m and 22 h p.i. Following [^64^Cu]Cu-DOTA administration, PET/CT acquisitions were performed at 40 min, 2 h and 6 h p.i.

### 4.9. Urine Sampling for Excretion Profiling

Urine samples were collected from two patients at irregular timepoints after the intrathecal administration of [^64^Cu]Cu-DOTA. Sampling was performed between 1.75 h and 24 h p.i. In total, seven samples were taken from patient 1 and six from patient 2. The total volume of each urine sample was determined. The samples were homogenized by shaking prior to withdrawing an aliquot of 100 µL. The aliquot was transferred into a test tube and the activity in 100 µL was determined using a gamma ray spectrometer (Cryo-Pulse5 plus, Mirion Technologies (Canberra) GmbH, Hamburg, Germany). The total activity in each urine sample was calculated by multiplying the activity concentration by the total sample volume.

## 5. Conclusions

PET RNC with [^64^Cu]Cu-DOTA appears to be a feasible imaging procedure for identifying CSF leaks when CT and MRI have failed, leave doubt regarding the findings, or are not applicable due to contraindications. Due to the very low content of excipients, the safety profile of [^64^Cu]Cu-DOTA must be considered superior to CT or MRI contrast agents and even to some formulations of [^111^In]In-DTPA or [^99m^Tc]Tc-DTPA. Compared to RNC with [^111^In]In-DTPA, PET RNC with [^64^Cu]Cu-DOTA provides images of decidedly higher spatial resolution, allowing for the identification of even very small CSF leaks.

## Figures and Tables

**Figure 1 pharmaceuticals-16-01269-f001:**
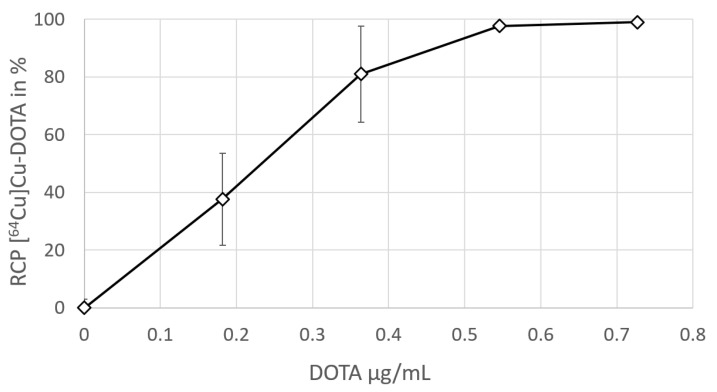
Percentage of [^64^Cu]Cu-DOTA in the reaction mixtures (PBS, 550 µL, pH = 7.4–7.5, 100 °C, 15 min) containing varying concentrations of DOTA. The percentage of [^64^Cu]Cu-DOTA was determined via radio TLC.

**Figure 2 pharmaceuticals-16-01269-f002:**
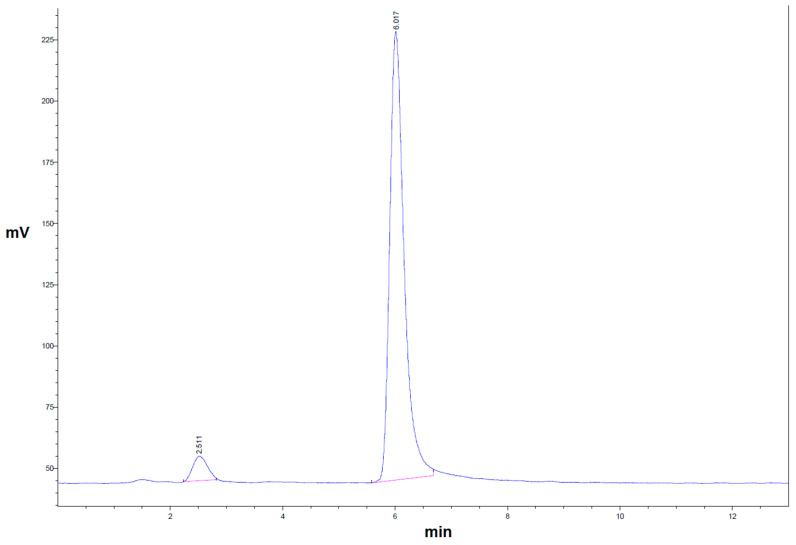
Representative radio HPLC chromatogram showing [^64^Cu]Cu-DOTA as a single peak at a retention time of 6.0 min, with an RCP of 94.8%, while non-labeled [^64^Cu]Cu-chloride (5.2%) is detected at 2.5 min.

**Figure 3 pharmaceuticals-16-01269-f003:**
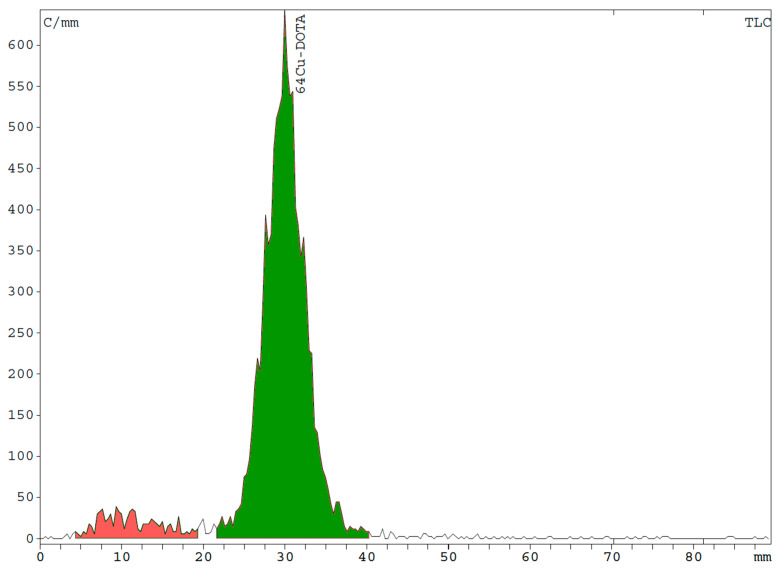
Representative radio TLC chromatogram showing [^64^Cu]Cu-DOTA at R_f_ = 0.3 with an RCP of 92.9%, while non-labeled [^64^Cu]Cu-chloride is detected at the bottom of the TLC plate at R_f_ = 0.0–0.1.

**Figure 4 pharmaceuticals-16-01269-f004:**
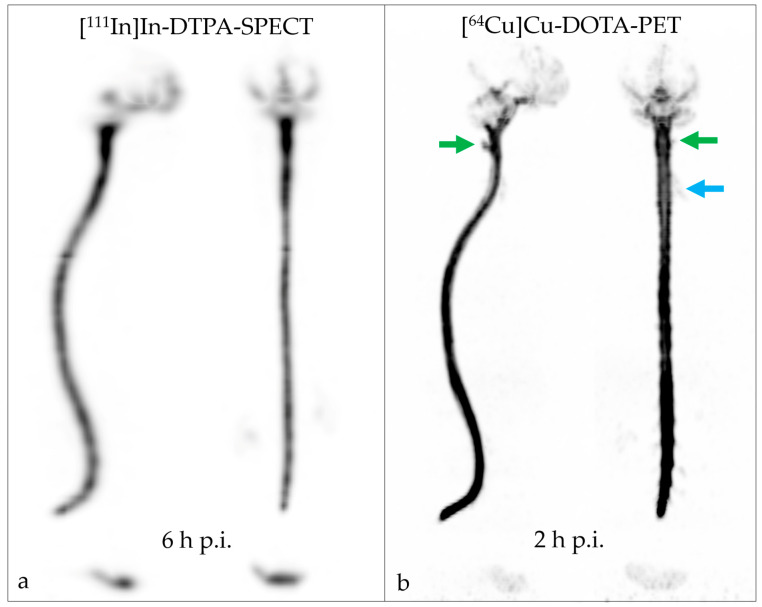
[^111^In]In-DTPA (**a**) and [^64^Cu]Cu-DOTA PET (**b**) of a patient experiencing symptoms of chronic CSF loss. Using PET, a CSF leak between the vertebral bodies C1 and C2 was identified (green arrows), along with visible tracer distribution along the left C5 nerve root sleeve (blue arrow). The higher image resolution of PET in comparison to SPECT facilitates CSF leak identifications.

**Figure 5 pharmaceuticals-16-01269-f005:**
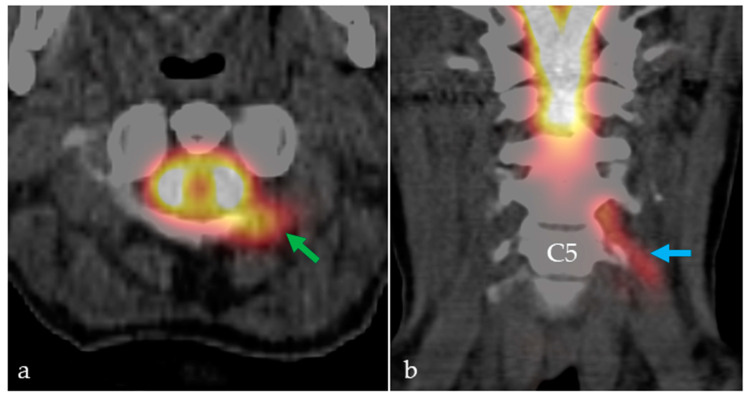
[^64^Cu]Cu-DOTA PET/CT fusion images (2 h p.i.) clearly show the atlantodental CSF leak ((**a**), axial view, green arrow). Furthermore, activity was seen in the sleeve of the left C5 nerve root sleeve ((**b**), coronal view, blue arrow), which may represent a normal variant.

**Figure 6 pharmaceuticals-16-01269-f006:**
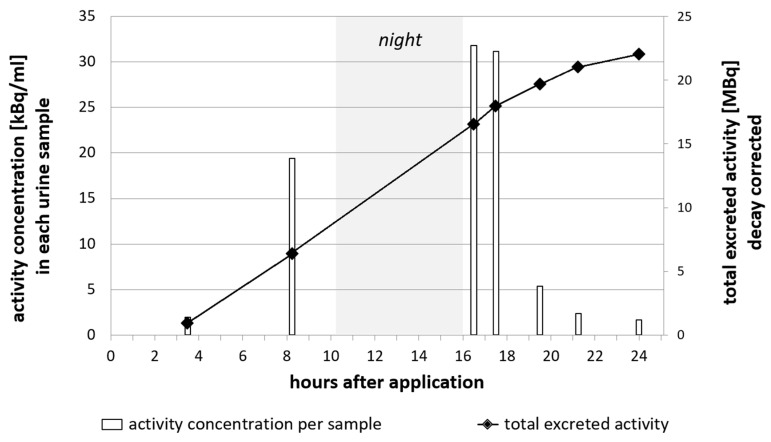
Excretion profile of [^64^Cu]Cu-DOTA in a patient with a spinal CSF leak 24 h after intrathecal administration. Columns indicate the activity concentration per sample. The marked line denotes the increasing sum of total excreted activity.

**Table 1 pharmaceuticals-16-01269-t001:** Percentage of [^64^Cu]Cu-DOTA in the reaction mixtures, using starting amounts of 60 ± 5 MBq [^64^Cu]Cu-chloride in PBS (5 mL, 100 °C, 15 min).

Batch No.	Starting Amount of DOTA in µg	Starting Amount of [^64^Cu]Cu Chloride in µL	RCP of [^64^Cu]Cu DOTA in %	pH of Reaction
1	2	120	27	7.3
2	10	120	86	7.4
3	20	120	96	7.3
4	20	400 ^1^	99	6.8

^1^ Batch no. 4 was produced on the day after delivery, resulting in a higher volume of [^64^Cu]Cu-chloride (400 µL) being required to achieve a starting activity of 60 ± 5 MBq.

**Table 2 pharmaceuticals-16-01269-t002:** Quality control specifications of [^64^Cu]Cu-DOTA. Results are given as the mean ± SD (n = 12).

		**Product Specifications**	**[^64^Cu]Cu-DOTA**
		activity	57.5 ± 1.6 MBq
		radiochemical yield	96 ± 3%
		volume	10.5 ± 0.7 mL
		specific activity	1.2 ± 0.1 MBq/µg
		activity concentration	5.6 ± 0.2 MBq/mL
**Quality Control**	**Method**	**Acceptance** **Criteria**	**Result**
appearance	visual inspection	clear, colorless solution	complies
pH	potentiometric determination	7.0–7.6	7.4 ± 0.2
osmolality	osmolality determination	200–300 mOsm/kg	282 ± 7 mosm
radionuclide identity	gamma ray spectrometry	511 keV and 1345 keV	complies
radionuclidic purity ^64^Cu		≥99.5%	99.9 ± 0.0001%
radiochemical purity	radio HPLC	[^64^Cu]Cu-DOTA ≥ 90.0%	94.1 ± 2.3%
radiochemical purity	radio TLC		94.2 ± 2.5%
bacterial endotoxins	LAL test	≤1.0 IU/mL	≤0.5 ± 0.0 IU/mL
sterility	sterility testing	sterile	complies

**Table 3 pharmaceuticals-16-01269-t003:** Composition and specifications of [^64^Cu]Cu-DOTA as compared to [^111^In]In-DTPA (Curium and GE Healthcare) and [^99m^Tc]Tc-DTPA.

	[^64^Cu]Cu-DOTA	[^111^In]In-DTPA [16,18]	[^99m^Tc]Tc-DTPA [15,21]
total volume of formulation	10 mL	0.5 mL or 1.0 mL	5.0 mL, depending on kit
max. volume per administration	5 mL	1.0 mL	0.5 mL
activity concentration	5.6 ± 0.2 MBq/mL	37 MBq/mL	560 MBq/mL
activity per administration	30 MBq	Curium: 18.5–37 MBq GE Healthcare: 18.5	280 MBq
pH	7.0–7.6	Curium: 7.0–7.6 GE Healthcare: 7.0–8.0	4.0–5.0, ≥6.0 after buffering with NaOH
expiration time	24 h	Curium: 24 h GE Healthcare: 7 d	6 h
**max. content of excipients per administration, assuming max. volume per administration**			
DTPA	--	Curium: 0.1 mg or 0.25 µmol GE Healthcare: 50 µg or 0.13 µmol	up to 25 mg (50 µmol CaNa_3_DTPA) per kit, max. 1 mg (2 µmol) recommended per administration
DOTA	25 µg (0.06 µmol)	--	--
calcium(II) ion	9 µg CaCl_2_ (0.06 µmol)	Curium: 37.4 µg or 0.25 µmol CaCl_2_ × 2H_2_O GE Healthcare: none	equimolar amounts to DTPA due to being present in the chemical formula of CaNa_3_DTPA
sodium hydroxide	--	Curium: 0.73 mg GE Healthcare: --	unspecified content
sodium chloride	40.0 mg or 0.68 mmol	Curium: 7.9 mg or 0.14 mmol GE Healthcare: --	4.5 mg or 0.08 mmol (assuming 0.9% saline used for reconstitution)
phosphate ion	0.092 mmol (11.9 mg or 0.085 mmol Na_2_HPO_4_ and 1 mg or 0.007 mmol KH_2_PO_4)_	Curium: 5.0 mg or 0.014 mmol Na_2_HPO_4_ × 12H_2_O GE Healthcare: --	--
sodium bicarbonate	--	Curium: -- GE Healthcare: content verified but concentration not specified	
water	5000 mg	897–997 mg	
other excipients	--	--	SnCl_2_ × 2H_2_O (max. 0.5 mg) depending on kit: gentisic acid, sodium *p*-amino-benzoic acid

## Data Availability

Data sharing is not available for this article.

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
