# Peer review of "Radionuclide Cisternography with [64Cu]Cu-DOTA"

_pharmaceuticals, 2023, doi:10.3390/ph16091269_

Round 1

Reviewer 1 Report

This manuscript describes the GMP production of [64Cu]Cu-DOTA including a case report of PET-cisternography comparing with the SPECT method.

The manuscript is well written and the GMP-production and quality controls follows the guidelines for radiopharmaceuticals. 

The PET-image clearly show the advantage over SPECT and the clinical value of [64Cu]Cu-DOTA justify publication of the described method.

Is there any hypothesis why excess DOTA (20ug) was needed?

On line 223 the authors use ...non-labeled [64Cu]Cu-chloride....

maybe unconjugated or non-chelated could be used. 

Just two editorial details which could be considered. Acccording to guideline for radiotracers fluorine-18 is preferred over 18F in text (line 116) and "cold" on line 239 should not be used and thus be deleted. 

Reviewer 2 Report

The authors report on a study using a known radiometal complex, [64Cu]Cu-DOTA, as an alternative for PET radionuclide cisternography (RNC).

The introduction provides an excellent background about the state of the art about (radio)pharmaceuticals in the clinical setting, their limitations, as well as the importance of accurate tracers for RNC detection.

The experimental section is very detailed and addresses the main key points about GMP production as well as in vitro and potential in vivo instability.

Imaging studies in patients reveal that [64Cu]Cu-DOTA  is suitable for the detection of CSF leaks and is superior regarding image resolution compared to conventional SPECT agents. Biodistribution studies show that the radiotracer is excreted via the urinary tract and small levels of radioactivity accumulate in the bladder, and in the liver after 24h p.i..

The authors address formulation and dose issues as well as the advantages of 64Cu]Cu-DOTA for RNC imaging. the authors also highlight that no adverse effects were observed. 

The authors should comment on the reason why images of only one patient are used to support the potential of the radiotracer. 
